**Subject Category:**
Biology (whole organism)

genetics/molecular biology/environmental science

endochitinase, entomopathogenic fungi, *Bemisia tabaci*, chitinase activity, insect resistance, virus-induced gene silencing

**Author for correspondence:**
Adnan Akhter
e-mail: adnanakhter.iags@pu.edu.pk

# Chitinase genes from *Metarhizium anisopliae* for the control of whitefly in cotton

Waheed Anwar[1], Muhammad Asim Javed[1], Ahmad Ali Shahid[1,2], Kiran Nawaz[1], Adnan Akhter[1], Muhammad Zia Ur Rehman[1], Usman Hameed[1], Sehrish Iftikhar[1] and Muhammad Saleem Haider[1]

[1]Institute of Agricultural Sciences, and [2]Centre of Excellence in Molecular Biology, University of the Punjab, Lahore, Pakistan

 AA, 0000-0001-6482-2116

Entomopathogenic fungi produces endochitianses, involved in the degradation of insect chitin to facilitate the infection process. Endochitinases (*Chit1*) gene of family 18 glycosyl hydrolyses were amplified, cloned and characterized from genomic DNA of two isolates of *Metarhizium anisopliae*. Catalytic motif of family 18 glycosyl hydrolyses was found in *Chit1* of *M. anisopliae*, while no signal peptide was found in any isolate, whereas substrate-binding motif was found in *Chit1* of both isolates. Phylogenetic analysis revealed the evolutionary relationship among the fungal chitinases of *Metarhizium*. The *Chit1* amplified were closely related to the family 18 glycosyl hydrolyses. Transient expressions of *Chit1* in cotton plants using Geminivirus-mediated gene silencing vector of *Cotton Leaf Crumple Virus* (CLCrV) revealed the chitinase activity of *Chit1* genes amplified from both of the isolates of *M. anisopliae* when compared with the control. Transformed cotton plants were virulent against fourth instar nymphal and adult stages of *Bemisia tabaci* which resulted in the mortality of both fourth instar nymphal and adult *B. tabaci*. Thus, the fungal chitinases expressed in cotton plants played a vital role in plant defence against *B. tabaci*. However, further studies are required to explore the comparative effectiveness of chitinases from different fungal strains against economically important insect pests.

## 1. Introduction

The whitefly (*Bemisia tabaci*) is recognized as an economically important insect pest of crop plants causing damage in both field and greenhouse through feeding on phloem sap, excretion

of honey dews and by transmitting viruses [1–4]. Major damage caused by *B. tabaci* is associated with its notable property of transmitting numerous viruses. Brown [5] reported 1100 whitefly species worldwide; however, only three among them, named as *B. tabaci*, *Trialeurodes vaporariorum* and *T. abutiloneus*, were known as the vector of plant viruses [5]. However, *B. tabaci* (Gennadius) is considered the most significant of the whitefly vectors with the ability to transmit geminiviruses [6]. There are about 288 species of begomoviruses (family: Geminiviridae) which are known to be transmitted by *B. tabaci* [7].

Different strains of *M. anisopliae* are extensively used as possible biological insect control agents [8] against *B. tabaci*, coconut beetle, termite, grasshoppers and rice bug termite [9–11]. In recent years, several studies have suggested that the entomopathogenic fungi producing chitinase may be used as an effective organic control agent against whitefly. The insect cuticle is fabricated of crystalline chitin nanofibres in proteins, lipids matrix and polyphones that provide a physical fence against pests and diseases [12]. For the biological control of various insects, microbial chitinolytic enzymes are considered very significant in this regard as they have the ability to impede chitin evidence [8]. Fungal chitinases have the potential to serve as a means of expansion of biocontrol agents of insects. Habeeb *et al.* [13] recorded up to 100% mortality rate of the *Hyalomma dromedarii* eggs by using a mixture of protease and chitnase enzymes originating from the soil fungi. However, the biocontrol activity of different isolates of *Metarhizium anisopliae* was found to be dependent on the changing osmotic conditions [14].

Chitin, a poly-β-1, 4-*N*-acetylglucosamine, is one of the main structural component of the cell wall of various plant pathogenic fungi as well as of cuticle and shells of arthropods, insects, molluscs and crustaceans [15,16]. Chitin consists of 22–44% of the cell wall substance of fungi and it is crucial for preserving the systemic reliability of hyphae [17]. Gravimetric investigation showed that the chitin substance comprises up to 40% of the exuvial parches mass relying on the insect classes [18]. Chitin is formed in the exo and endocuticle section of the integument [19]. Chitinases from insect-associated fungi can be used to develop transgenic plant and increase fungus virulence against whitefly. The entomopathogenic fungus *M. anisopliae* genome encodes different chitinases. Fungus strains, when over-expressing the *Chit2* gene, increase the killing efficiency towards its prey. While the fungal strains that lack gene encoding *Chit2* showed decreased pathogenicity towards insects. *Metarhizium anisopliae* encode *Chit2* gene causes pathogenicity of transgenic isolates through silencing and overexpression of the chitinase gene [20].

The role of chitinase in plant defence against fungal attack has been very well documented [21,22]. Genes encoding for fungal and plant chitinases are very efficient at controlling a large number of plant pathogenic fungi [23–24]. Several classes of genes have been used in genetic engineering to develop resistance in cotton to fungal pathogens, but there are no data available concerning the transgenic approach of chitinase genes against insects. In the last two decades, efforts have been made in the transgenic expression of plant fungal chitinase in crop transgenic plants [25]. Therefore, the present study was designed with the following objectives: (i) isolation and characterization of chitinase genes from different isolates *M. anisopliae*, (ii) transient expression of chitinase genes in cotton plants by using virus-induced gene silencing (VIGS) vectors, and (iii) evaluation of virulence of chitinase transient cotton plants against *B. tabaci*. The anticipated results will help in estimating the biocontrol potential of fungal chitinases to develop resistant plants against different insect pests.

# 2. Methodology

## 2.1. Isolation of chitinase gene from *M. anisopliae*

Chitinase genes were amplified from different isolates of *M. anisopliae* isolated from cotton mealybug and *B. tabaci*. Genomic DNA was extracted through modified CTAB method [26] and PCR was performed using Meta_Chit1_comp forward (5′-TCCCATGTTCTGTACTCGTTC-3′) and reverse primers (5′-CCCTTGCTCTTGAGGTAGGTAAC-3′). For the PCR, a reaction mixture containing 18.4 µl nuclease-free water, 2 µl of 1× tag buffer, 1 µl of 10 mM dNTPs, 0.5 µl of 25 mM $MgCl_2$, 0.5 µl of each forward and reserve primer (conc. of 10 pmol in 1 µl), 0.1 µl of Tag polymerase (5 U µl$^{-1}$, Thermo Fisher Scientific) and 2 µl of genomic DNA as template (50 ng µl$^{-1}$) was prepared. The PCR condition was initial denaturation of 95°C for a minute, 35 cycles at 95°C for a minute as denaturation time followed by annealing temperature of 55°C for 1 min for the isolation of *Chit1* genes from *M. anisopliae*. The extension time for PCR reactions was 2 min at 72°C. Finally, the reactions were extended for 10 min at 72°C.

## 2.2. Cloning in pGEM T-easy vector

PCR products were purified from gel using a Microgen purification kit according to the protocol provided by the manufacturer. Purified PCR products were ligated into pGEM T-easy cloning vector and transformed into DH5α strain of *Escherichia coli*. Cloned plasmids were isolated and confirmed through restriction with *EcoRI* before sequencing.

## 2.3. Sequencing and homology analysis

The plasmids were sent for sequencing to Eton Biosciences, San Diego, CA, USA, with M13 forward and reverse primers. Sequences were assembled for forward and reverse strands using DNA SeqMan pro software (https://www.dnastar.com/t-seqmanpro.aspx) and vector sequence was trimmed from the start and end. The final sequences were blast on NCBI nucleotides database.

## 2.4. Characterization of chitinase genes

For characterization of chitinase genes, sequences were blast on NCBI database and their percentage homology and evolutionary relationship were studied. BLAST analysis (NCBI; http://www.ncbi/ BLAST; [27]) and ClustalW software (htpp://www.ch.embnet.org/software/ClustalW.html; [28]) were used to compare the *Chit1* sequence with other chitinases available in the public database. The ORFs were analysed to find the conserved domain using the Conserved Domain Search service (CD-Search https://www.ncbi.nlm.nih.gov/Structure/cdd/wrpsb.cgi) tool from NCBI. Molecular weight and signal peptide of isolated chitinase were predicted by using software SignalP 3.0 (http://www.cbs. dtu.dk/services/SignalP/; [29]).

## 2.5. Homology modelling and validation

The homology modelling approach was used to predict the three-dimensional model of chitinase. The deduced amino acid sequence of *Chit1* was submitted to the automated comparative protein modelling server (http://swissmodel.expasy.org//SWISS-MODEL.html). Five chitinases templates (1D2 K, 1LL7, 1LL6, 1WNO, 1W9P) for *M. anisopliae* with 99% identity to *Chit1* amino acid sequence were selected for modelling. The deduced amino acid sequences for these queries were subjected to a BLAST search (BLASTp, using default parameters) against a Protein Data Bank (PDB) database [30]. The selected homology model was evaluated for compatibility with various structural parameters and stereo-chemical properties using reliability and comparative assessment tools PROCHECK [31]. Structure quality was determined in term of residues % in favourable, glycine residue and non-proline %, etc.

## 2.6. Expression of chitinase open reading frames in cotton plants

### 2.6.1. Virus-induced gene silencing vector

A VIGS vector derived from *Cotton Leaf Crumple Virus* (CLCrV) was obtained from a permanent inventory of clones from Brown's Lab, School of Plant Sciences, University of Arizona, Tucson, AZ, USA, developed specifically for transient gene expression for cotton [32]. In this VIGS vector, the ORF coding coat protein of component A of *Cotton Leaf Crumple Virus* (pJRTCLCrVA 008) was replaced with Multiple Cloning Site (MCS) (Genbank EU541443). The B component of *Cotton Leaf Crumple Virus* was also obtained from Brown's Lab, University of Arizona, AZ, USA.

### 2.6.2. Isolation of chitinase open reading frames

For the isolation of open reading frames (ORFs) containing chitin catalytic domain, primers were designed containing *NheI* restriction site in reverse primers and *EcoRI* in forward primers. ORFs from *M. anisopliae* isolates were amplified using Met_ORF forward (5′-TGGGAATTCATGAAGACGATGTTGTCTATTGG-3′) and reverse primer set (5′-GTAGCTAGCGAATTCACTAATAAGTTCTTGGG-3′) from previously cloned chitinases in the pGEM T-easy cloning vector. The PCR condition with the initial denaturation temperature of 95°C for a minute, 35 cycles at 95°C for a minute as denaturation time followed by different annealing temperature for different ORF of *M. anisopliae* were used. The annealing temperature

was 54°C for *Chit1* gene of *M. anisopliae*. The extension time for PCR reactions was 2 min at 72°C. Finally, the reactions were extended for 10 min at 72°C.

### 2.6.3. Construction of VIGS-Chit recombinant plasmids

The ORFs of *M. anisopliae* were ligated into pGEM T-easy vector and transformed into DH5α strain of *E. coli*. Plasmids were isolated, sequenced and analysed before cloning into VIGS vector. Plasmids and VIGS vector (CLCrV-A) were restricted with *EcoRI* and *NheI* restriction enzymes to clone into VIGS vector. Already restricted ORF of *Chit1* was ligated into VIGS vector at the ratio of 3:1 (insert : vector), transformed into DH5α strain of *E. coli*. Recombinant plasmids were isolated and confirmed by restriction analysis and sequencing of VIGS-Chit recombinant plasmids.

### 2.6.4. Particle bombardment

Seeds of cotton (*Gossypium hirsutum*) plant were germinated in 7-inch square pots filled with potting soil. Four seeds were used in the single pot and seedlings were grown for 5–10 days until the first true leaves appeared. Leaves were inoculated biolistically, transferred to the individual pot and kept in the growth chamber with a photoperiod of 16/8 h at 900 µmol m$^{-2}$ s$^{-1}$ and a temperature of 23–25°C. The experiment comprises five replicates, with each replicate representing a pot with four seedlings. Miracle Gro (Miracle Gro Products, Inc.) fertilizer was used twice a week and temperature comparison experiments were carried out in the Institute of Agriculture, University of the Punjab, by using 30°C/26°C or 22°C/18°C day/night temperature cycles, relative humidity 40–50%. Plants received ambient day length with 3 h during the dark period with 11–12 µmol m$^{-2}$ s$^{-1}$ of luminous light to activate day-long responses during the year. Plants were started independently in 225 ml styrofoam cups with one-third peat-lite (WR Grace Co.) and two-thirds pea gravel, potting mixture and transplanted to 1650 ml pots 2 dpi. Fertilizers and water were applied three times during the week with weak Hoagland solution [33]. Gold microprojectiles of 1 µm diameter (InBio) coated with a mixture of 5 µg each of the A (containing *Chit1*) and B components of CLCrV were used for the bombardment of seedlings [34]. DNA-coated micro-projectiles were loaded onto the filter end of a Millipore Swinnex filter and each seedling was placed directly under the outlet and bombarded once. Four seedlings were shot one plant at a time with an outlet pressure of 30–60 psi, with 60 psi becoming the standard for routine work except for the temperature comparison experiment. Seedlings were inoculated by using a commercially available Particle Delivery System (BioRad PDS1000-He) with 0.5 µg of each component, causing an average infection rate of 70%.

### 2.6.5. Transgenic expression analysis

Total RNA was extracted from transformed cotton leaves according to the method described by Kjemtrup *et al*. [35]. The PCR was performed at 25°C for 10 min, 37°C for 120 min and 85°C for 5 min followed by 35 cycles. For the confirmation of expression analysis, total RNA was extracted from young cotton leaves after 14 days and cDNA was synthesized. The coat protein region of *Cotton Leaf Crumple Virus* was amplified using (AGTTCTAGAATCACCTTCCACTATGAGAC) forward primer and (TCAGAATTCCCT TAACGTGCGATAGAT TCTGGGC) reverse primers [32]. The PCR products were run on 1% agarose gel for confirmation of transient expression.

## 2.7. Chitinase assay

Chitinase activity was measured using a chitinase assay kit (CS0980 Sigma-Aldrich). The fluorometric assay is based on the enzymatic hydrolysis of the chitinase substrate. *p*-Nitrophenol is released upon hydrolysis and can be measured fluorometrically at 405 nm. One unit will release 1.0 µmol of *p*-nitrophenol from the substrate per minute at 37°C. The substrate used for endochitinase was 4-Nitrophenyl β-D-*N,N′,N″*-triacetylchitriose. Purified chitinase provided in the kit was used as a positive control.

## 2.8. Virulence bioassay of chitinase transgenic plants against *B. tabaci*

After confirmation of expression of different chitinase in plants through VIGS and chitinase enzyme assay, the fourth instar nymph and adult *B. tabaci* were allowed to feed on cotton plants showing transient expression, and data after different intervals regarding percentage mortality were calculated using the modified Abbot's formula [36].

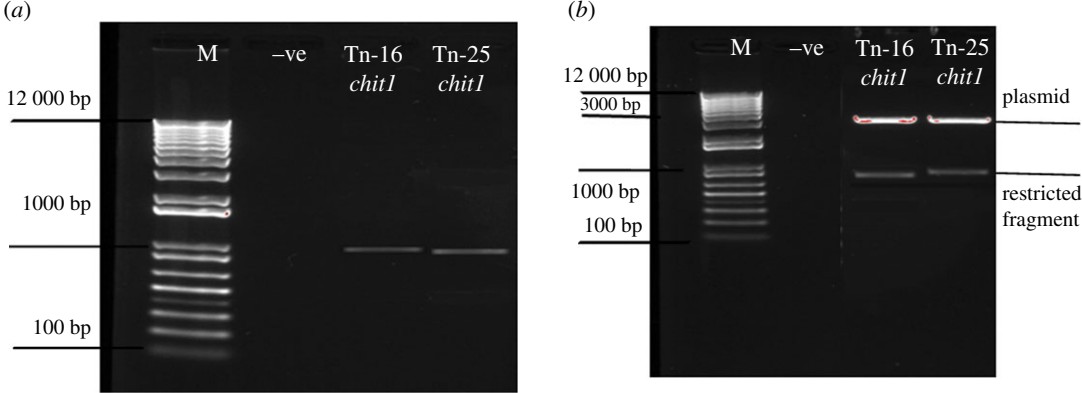

**Figure 1.** PCR confirmation (*a*) of partial endochitinase *Chit1* from *M. aniopliae* and restriction analysis (*b*) of transformed plasmid by using *EcoR1* restriction enzyme. M represents Promega™ plus 1 kb DNA ladder.

## 2.9. Data analysis

The data were analysed using SPSS, v. 11.5 statistical software (SPSS Inc., Chicago, IL, USA). The percentage data were transformed before analysis. Means were separated by Tukey's HSD at $p \leq 0.05$ level of significance.

# 3. Results

## 3.1. Isolation of endochitinase *Chit1*

The chitinase gene was isolated from the genomic DNA of *M. aniopliae* and PCR reaction produced an expected size (approx. 1 kb) bands for *Chit1* from *M. anisopliae* (figure 1).

The partial endochitinase *Chit1* recombinant clones were confirmed through restriction digestion (figure 1) and sequenced using M13 primers employing the primer walking technique. Vector sequences were removed from the nucleotide sequence through GENE TOOL and subjected to BLAST analysis. It showed that the nucleotide sequence of *Chit1* had maximum similarity to endochitinases compared to other entomopathogenic fungi. The *Chit1* showed 93% homology to endochitinases from *M. anisopliae Chit1* (AF027498).

## 3.2 Characterization of endochitinase *Chit1* from *M. anisopliae* isolate Tn-16 and Tn-25

The endochitinase gene of *M. anisopliae* isolate Tn-16 and Tn-25 was composed of 299 and 225 amino acids, respectively. The endochitinase gene of both isolates encoded a mature protein *Chit1* and lacked signal peptide. The encoded protein of Tn-16 has a molecular mass of 32.48 kDa and is basic in nature with a pI value of 8.43 (table 1), whereas the encoded protein of Tn-25 has a molecular mass of 27.73 kDa and is acidic in nature with a pI value of 5.05 (table 1). Conserved Domain search analysis revealed that *Chit1* consists of two domains, a chitin-binding domain and a catalytic glycosyl hydrolase family 18 (GH18) domain. The chitin-binding domain was located at C-terminal to the catalytic domain. It was evaluated on the basis of sequence alignment that structural motifs that were responsible for substrate-binding SIGG (SxGG) and catalytic motif DGIDVDWEYP (DxxDxDxE) were highly conserved in *Chit1* (shown in black box in electronic supplementary material, figure S1). The potential substrate-binding site of the Tn-16 isolate was placed at the 60 aa site and the catalytic active site was positioned at the 97 aa site. While potential substrate-binding site of Tn-25 was placed at the 6 aa site and the catalytic active site was positioned at the 43 aa site. BLAST analysis showed that the sequence of *Chit1* revealed 98% identity with the amino acid sequence of *M. anisopliae* strain (KJK95695 and ACU30519).

## 3.3. Phylogenetic analysis of endochitinase of family 18 glycosyl hydrolases

Phylogenetic analysis was carried out to consider the evolutionary relationship between the chitinases of different biocontrol fungi. In the neighbour-joining tree, endochitinase of fungal isolates grouped into three different clades and showed maximum homology to fungal class 18 basic chitinases as

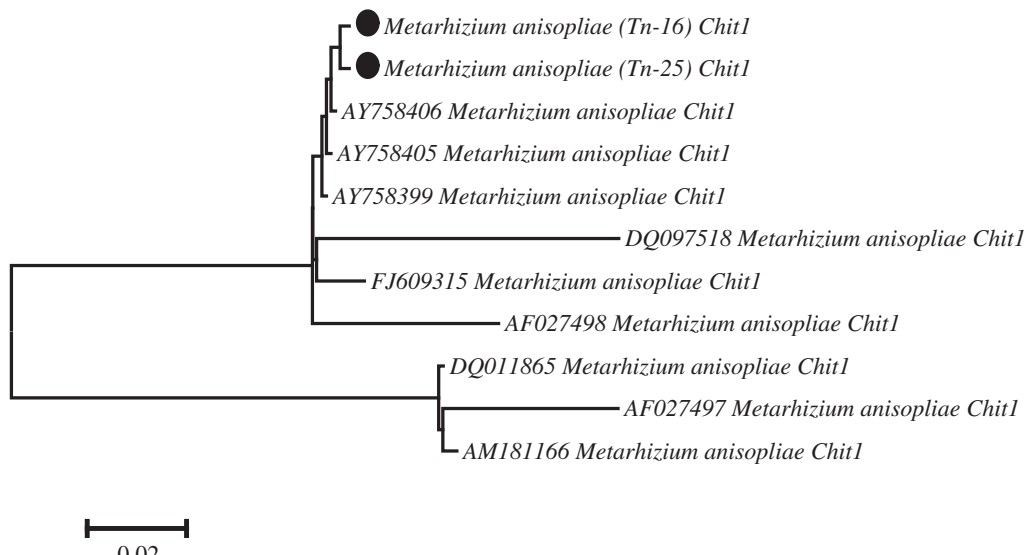

**Figure 2.** Phylogenetic analysis of endochitinase of family 18 glycosyl hydrolases isolated from entomopathogenic fungal species.

**Table 1.** Comparison of the active sites and amino acid residues of the family 18 endochitinases.

| species | total no. of residues | molecular wt. | pI value | signal peptides | residues of active sites | sequences of active site |
|---|---|---|---|---|---|---|
| *M. anisopliae* (Tn-16) | 299 | 32.48 kDa | 8.43 | — | 97–106 | DGIDVDWEYP |
| *M. anisopliae* (Tn-25) | 255 | 27.73 kDa | 5.05 | — | 43–52 | DGIDVDWEYP |

determined by a multidomain structure analysis. *M. anisopliae* isolates [Tn-16 (Genbank Accession No: MN218605) and Tn-25 (Genbank Accession no: MN218606)] of *Chit1* were classified into a distant clade and showed maximum similarities with *M. anisopliae Chit1* (AY758406 and AY758405). A comparative gene sequence analysis revealed that our endochitinase *Chit1* are the most closely related to family 18 glycosyl hydrolases as shown in figure 2.

## 3.4. Homology modelling and validation of *Chit1* and *Chit2* protein

Homology modelling showed that the *Chit1* of entomopathogenic fungi contain (α/β) 8 TIM barrel structure like other members of the class 18 hydrolase family. Generated models were validated by using Ramachandran plot calculations computed with the PROCHECK program. The homology model of the *Chit1* protein and its catalytic and binding domain motifs are shown in figure 3. Altogether 99–100% of the residues were in favoured and allowed regions. The Ramachandran plot of *M. anisopliae Chit1* (Tn-25) and *M. anisopliae Chit1* (Tn-16) showed 90.4, 91.5 and 8.7%, 7.7% residues in favoured and allowed regions, respectively. The Φ and Ψ distributions of the Ramachandran plots of non-Proline and non-Glycine residues are given in figure 4.

## 3.5. Confirmation of chitinase in cotton plants

The ORFs encoding conserved domain of catalytic family 18 from *Chit1* gene of *M. anisopliae* (Tn-25 isolate) were selected for their detection in cotton plant through VIGS vector modified using a component of cotton leaf crumple virus. The specific ORFs of chitinase were amplified from recombinant plasmids and amplicon of 900 bp was observed with ORFs-specific primer (figure 5). Restriction analysis and sequencing confirmed the VGS-*Chit1*_tn25 construct. Amplification was positive when confirmed by amplifying coat protein specific primers and ORFs-specific primers after 6 days inoculation. The symptoms of cotton leaf crumple virus also confirmed the expression of chitinase in cotton plants (figure 6).

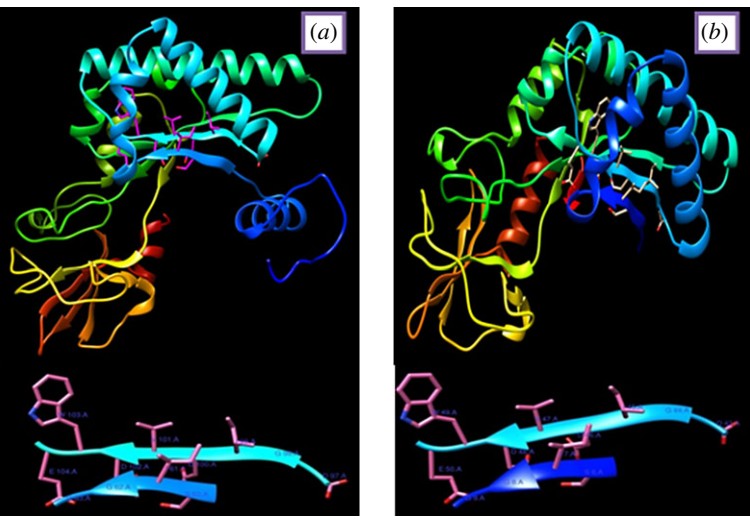

**Figure 3.** Homology modelling of *Chit1* protein of *M. anisopliae* (Tn-25, *a*) and *M. anisopliae* (Tn-16, *b*). The side chains of catalytic and binding domain motifs are shown here. The deduced amino acid residues of catalytic (DGIDVDWEYP) and substrate-binding site (SIGG) are indicated. The residues of catalytic domain are depicted in pink colour and sky blue ribbon, while the residues of binding domain are depicted in pink colour and dark blue ribbon.

(*a*)

Ramachandran plot statistics

| | no. of residues | %-tage |
|---|---|---|
| most favoured regions [A, B, L] | 215 | 91.5% |
| additional allowed regions [a, b, l, p] | 18 | 7.7% |
| generously allowed regions [~a, ~b, ~l, ~p] | 2 | 0.9% |
| disallowed regions [XX] | 0 | 0.0% |
| non-glycine and non-proline residues | 235 | 100.0% |
| end-residues (excl. Gly and Pro) | 1 | |
| glycine residues | 26 | |
| proline residues | 12 | |
| total number of residues | 274 | |

(*b*)

Ramachandran plot statistics

| | no. of residues | %-tage |
|---|---|---|
| most favoured regions [A, B, L] | 198 | 90.4% |
| additional allowed regions [a, b, l, p] | 19 | 8.7% |
| generously allowed regions [~a, ~b, ~l, ~p] | 2 | 0.9% |
| disallowed regions [XX] | 0 | 0.0% |
| non-glycine and non-proline residues | 219 | 100.0% |
| end-residues (excl. Gly and Pro) | 2 | |
| glycine residues | 23 | |
| proline residues | 11 | |
| total number of residues | 255 | |

**Figure 4.** The Ramachandran plot showing the $\Phi$–$\Psi$ torsion angles for all residues in most stable predicted three-dimensional conformation of chitinase protein: (*a*) *M. anisopliae* isolate Tn-16 and (*b*) *M. anisoplaie* isolate Tn-25.

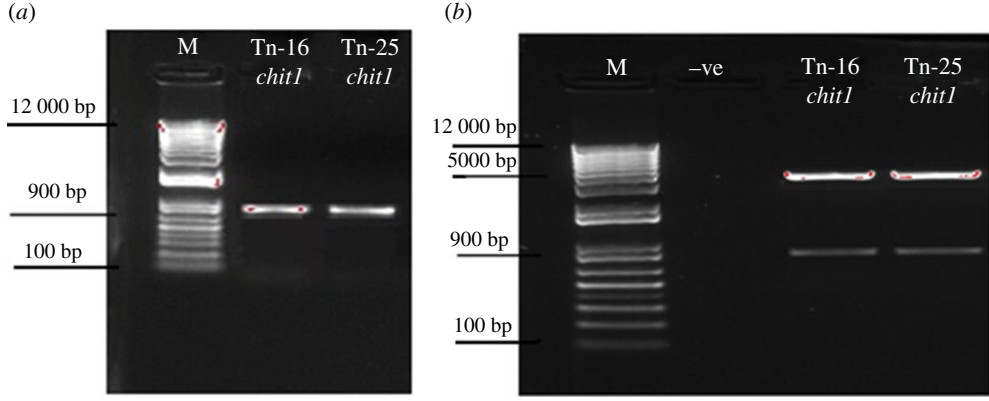

**Figure 5.** Amplification of chitinase ORFs (*a*) from *Metarhizium*. Restriction analysis (*b*) of VIGS-Chit recombinant plasmids. M represents Promega™ 1 kb Plus DNA ladder.

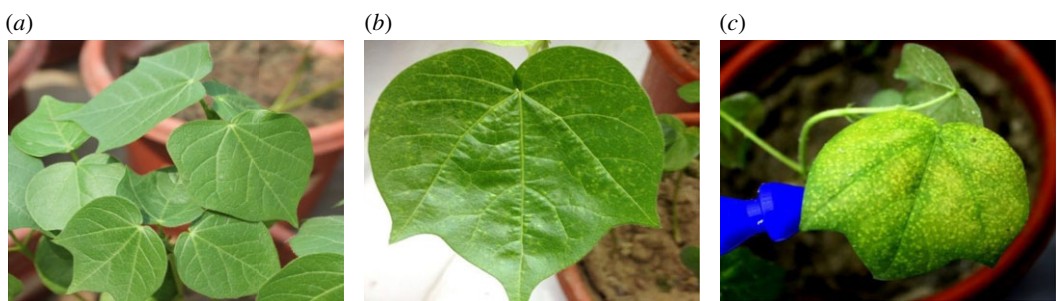

**Figure 6.** Visual evidence of CLCrV-induced gene silencing by using VIGS vector. Control (*a*), mock inoculated with empty VIGS vector and (*b*) cotton leaves showing expression of CLCrV-*Chit1* of *M. anisopliae* isolate (Tn-25) (*c*).

## 3.6. Chitinase activity assay

All transformed cotton plants produce chitinase including mock, control after 6 days. The transformed plants showed higher chitinase activity when compared with the control. Transformed plants of *Chit1* (Tn-25) of *M. anisopliae* showed activity of 5.55 U ml$^{-1}$. The net chitinase activity was calculated by reducing the amount of chitinase in the control. The activity of the control (VIGS + GUS gene) was 5.13 U ml$^{-1}$, while it was 5.09 U ml$^{-1}$ in mock as described in table 2.

## 3.7. Virulence bioassay of chitinase transformed plants against fourth instar nymph and adult stages of *B. tabaci*

The virulence of *Chit1* of *Metarhizium* (Tn-25) was promising and showed incomparable results. The mortality of the nymph started after 120 h of inoculation and was recorded as 18.37% after 6 days (table 3). The mortality of *B. tabaci* nymphs was recorded as 9.65% by inoculation of *Chit1* of *M. anisopliae* (Tn-25) after 6 days (table 3). The virulence of *Chit1* of *M. anisopliae* (Tn-25) was compared on the fourth instar nymphal and adult stages of *B. tabaci*. Nymphal stages of *B. tabaci* are more vulnerable when compared with the adults.

# 4. Discussion

Chitinases are particularly interesting as a part of the plant defence system because of their broad-spectrum fungicidal, insecticidal activity, while being non-toxic to plants, animals and higher vertebrates. Several genes encoding hydrolytic enzyme (endochitinase) were cloned and transferred to plants to impart resistance against plant pathogens [24,37,38]. Okongo *et al.* [39] reported the biocontrol activity of microbial chitinase against stem borers such as *Eldana saccharina*, as well as against the fungal pathogens Aspergillus, *Fusarium* and *Mucor* sp. In this study, the *Chit1* sequences isolated from entomopathogenic fungi belong to family 18 of glycosyl hydrolases based on amino acid similarity [24,40]. Chitin-binding domain (ChBD) is responsible for specific binding of chitinases to

**Table 2.** Chitinase activity of transformed cotton plants. $n = 5$.

| name of chitinase | total chitinase activity of transformed plants (U ml$^{-1}$) | chitinase activity in control (U ml$^{-1}$) | net chitinase activity (U ml$^{-1}$) |
|---|---|---|---|
| *Chit1* of *M. anisopliae* (Tn-25) | 5.55 | 5.09 | 0.46 |
| mock | 5.13 | 5.09 | 0.04 |

**Table 3.** Percentage mortality of chitinase transformed plants against fourth instar nymphal stage and adult of *B. tabaci*. $n = 5$, values with different letters showed significant difference at $p > 0.05$.

| sr. no | name of chitinase | mortality | after | | | | |
|---|---|---|---|---|---|---|---|
| | | | 48 h | 72 h | 96 h | 120 h | 144 h |
| 1 | *Chit1* of *M. anisopliae* (Tn-25) | nymphs | — | — | $8.75 \pm 1.60bc$ | $10.86 \pm 1.53b$ | $18.37 \pm 1.57a$ |
| 2 | *Chit1* of *M. anisopliae* (Tn-25) | adults | — | — | — | $3.44 \pm 1.69b$ | $9.65 \pm 1.61a$ |

insoluble chitin [41]. *Chit1* gene of *M. anisopliae var. anisopliae* encodes a chitinase with a similar molecular weight [42]. The structural motif of a substrate-binding site (SXGG) and catalytic domain motif (D1XXD2XD3XE) was recognized and found to be highly conserved in fungal chitinase of family 18 [43].

Homology modelling showed that the *Chit1* of entomopathogenic fungi contain the ($\alpha/\beta$) 8 TIM barrel structure similar to other members of the class 18 hydrolase family. The active site is formed by two signature sequences which lie along barrel strands 3 and 4 of the class 18 chitinases on the carboxyl end of the $\beta$-barrel and appear to be important both for the stability of the fold and for catalytic activity [44]. A neighbour-joining phylogenetic tree was constructed for the assessment of evolutionary relationships and classification of the predicted chitinase *Chit1* into fungal classes. In the neighbour-joining tree, endochitinase of fungal isolates grouped into three different clades and *M. anisopliae* isolates of *Chit1* classified into distant clade and gene sequence analysis revealed that endochitinase *Chit1* in the present study are most closely related to family 18 glycosyl hydrolases. Similarly, da Silva [45] also studied the evolutionary relationships of *Chit1* and the neighbour-joining tree was classified into five clades, *Metarhizium Chit30* and *Chit2* fell into the same clade and showed maximum homology with the plant-like chitinase. However, other clades were occupied with *Chit42* and shared similarity with fungal-like chitinase [46].

It is essential to analyse the functionality of the transgene product before transferring the gene into target crops. In order to assay the endochitinase activity (of transgene product) in transformed cotton, chitinase assay revealed that the *Chit1* from *M. anisopliae* (isolate Tn-25) has shown 0.46 U ml$^{-1}$ activity after 6 days. Previously, Nahar *et al.* [47] also reported the chitinase activity of 0.01–0.0398 U ml$^{-1}$ from various *M. anisopliae* isolates in culture media studied. The chitinolytic activity of 17 isolates of *M. anisopliae* was evaluated by Braga *et al.* [48] in different culture media and activity was found to vary from 0.0261 to 0.1340 U ml$^{-1}$. Kang *et al.* [49] reported higher chitinolytic activity (8.66 mU ml$^{-1}$) in the fluid culture when *M. anisopliae* were grown in a medium containing colloidal chitin as a sole carbon source. The enzyme activity significantly varied with variance in substrate and type of fungal isolate [50,51]. From our results, it is proposed that the entomopathogenic fungi isolates are virulent as they produced a greater amount of chitinase enzyme.

It has been well documented that greater activity of chitinase in plants can cause the reduction in the damage caused by pathogens [52]. Bioassay against target pathogen is the best way to learn the usability of the cloned gene in plants and all cotton plants produce mild symptoms of CLCrV such as downward leaf curling and lighter veins. Viral-induced gene silencing through geminivirus vectors in *Arabidopsis* [53] and *N. benthamiana* [35] produced severe symptoms when compared with tobacco rattle virus (TRV) VIGS vector that has been widely used [54,55]. The virulence of *Chit1* of *M. anisopliae* (Tn-25) showed 18.37% mortality of nymphs after 6 days of inoculation. When compared with nymphs, *Chit1*

of *M. anisopliae* (Tn-25) killed 9.65% adults of *B. tabaci* after 6 days of inoculation. The insect bioassay results displayed that overproduction of the novel endochitinase *Chit1* can increase the infection efficiency of *M. anisopliae* and promote infection against adults of *B. tabaci*.

Transformations of various plants with chitinase genes showing enhanced disease resistance has been achieved. Development of the Colorado potato beetle can be inhibited by the expression of chitinase in tomato [21]. Similarly, Kitajima *et al*. [56] also reported strong insecticidal activities associated with two chitinase proteins from the latex of Mulberry (*Morus* sp.) against the larvae of *Drosophila melanogaster*, while Suganthi *et al*. [57] documented 100% mortality of a tea mosquito bug (*Helopeltis* spp.) by using a chitinase from *Pseudomonas fluorescens*, and Fang *et al*. [58] evaluated that the overexpression of *B. bassiana* chitinase, gene *BbChit1* enhances the virulence against aphids. In future, the combination of chitnases with nanotechnology for the development of biopesticides will bring revolutionary changes in the agriculture sector [59].

# 5. Conclusion

It is concluded that fungal chitinases are diverse and effective against different stages of insect pests. The chitinase from *M. anisopliae* can be incorporated into cotton plants to enhance their resistance against *B. tabaci*. In addition to that, chitnases have the potential to suppress the phyto-pathogenic activity which in turn will significantly reduce the dependency of the farming community on toxic chemical plant protection strategies. However, there is a need to conduct broad-spectrum studies to assess the influence of chitnases on biodiversity by evaluating the pathogenicity of chitinases against beneficial insects before their consideration as biopesticides.

Ethics. For the collection of insect samples, prior permission was sought from the farm owners and facilitated by the Institute of Agricultural Sciences, University of the Punjab, Lahore, Pakistan.

Data accessibility. The genomic data generated in this study have been deposited in GenBank (accession nos.: MN218605–MN218606). The data available as electronic supplementary material contain details about nucleotides along deduced amino acids of Met_Chit1 gene of *M. anisopliae* isolate Tn-16 and Tn-25 as well as about the raw data for tables 2 and 3.

Authors' contributions. W.A. and A.A. perceived and designed the experiments. W.A., K.N., M.A.J. and M.Z.U.R. performed the experiments. M.S.H., A.A.S., A.A. and U.H. contributed with reagents, designing methodology and finalizing analysis tools. W.A., M.A.J. and S.I. analysed the data. A.A., W.A. and K.N. wrote and revised the paper scientifically and grammatically.

Competing interests. The authors declare that they have no competing interest.

Funding. This project was sponsored by the Higher Education Commission of Pakistan and Institute of Agricultural Sciences, University of the Punjab, Lahore, Pakistan.

Acknowledgements. The authors are thankful to the Dr Javed Iqbal and Dr Sajid Ali for the facilitation in molecular data and statistical data analysis, respectively. We are also grateful to the lab-technicians of Institute of Agricultural Sciences, for providing the technical assistance during various assays related to the present study.

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
