## [Reviewer comments · Royal Society Open Science]

Review History

RSOS-190412.R0 (Original submission)

Review form: Reviewer 1 (Muhammad Rana Kashif)

Is the manuscript scientifically sound in its present form?

Yes

Are the interpretations and conclusions justified by the results?

Yes

Is the language acceptable?

Yes

Is it clear how to access all supporting data?

Yes

Do you have any ethical concerns with this paper?

No

Have you any concerns about statistical analyses in this paper?

No

Recommendation?

Accept with minor revision (please list in comments)

Comments to the Author(s)

The manuscript titled "Chitinase genes from *Metarhizium anisopliae* for the control of Whitefly in Cotton" adds up valuable research output to plant pathology. The study appears to be authentic and may play significant role to potentially control insect pests causing spread of devastating viral diseases by employing chitinase genes from fungal strain *M. anisopliae*.

Following are minor comments to be considered,

Methodology part:

2.6.4. clarify that the number of seedlings and pots were set up in the experiment.

Results section.

Figures 2 & 3, can be supplementary files.

3.5. Expression of chitinase in cotton plants. It is obvious that the genes were detected based on end-point PCR and the data was validated by phenotypic expression. Since real expression of genes by real-time quantitative has not been done, therefore, here expression may be replaced by detection and confirmation of chitinase in cotton plants.

Review form: Reviewer 2 (Yilmaz Kaya)

Is the manuscript scientifically sound in its present form?

Yes

Are the interpretations and conclusions justified by the results?

No

Is the language acceptable?

Yes

Is it clear how to access all supporting data?

Yes

Do you have any ethical concerns with this paper?

No

Have you any concerns about statistical analyses in this paper?

No

Recommendation?

Accept with minor revision (please list in comments)

Comments to the Author(s)

Dear Authors;

Whiteflies are small Hemipterans that typically feed on the undersides of cotton leaves. They comprise the family Aleyrodidae, the only family in the superfamily Aleyrodoidea. More than 1550 species have been described.

In warm or tropical climates and especially in fields, whiteflies present major problems in cotton protection. Worldwide economic losses are estimated at hundreds of millions of dollars annually for cotton business.

Therefore, the present study was designed for the harmful effects with using genetically modified technology. This is an original research with finding results. I am strongly suggested that "accepted with minor correction".

Decision letter (RSOS-190412.R0)

13-Jun-2019

Dear Dr Akhter

On behalf of the Editors, I am pleased to inform you that your Manuscript RSOS-190412 entitled "Chitinase genes from *Metarhizium anisopliae* for the control of Whitefly in Cotton" has been accepted for publication in Royal Society Open Science subject to minor revision in accordance with the referee suggestions. Please find the referees' comments at the end of this email.

The reviewers and handling editors have recommended publication, but also suggest some minor revisions to your manuscript. Therefore, I invite you to respond to the comments and revise your manuscript.

- Ethics statement

- Data accessibility

If you wish to submit your supporting data or code to Dryad (<http://datadryad.org/>), or modify your current submission to dryad, please use the following link:
<http://datadryad.org/submit?journalID=RSOS&manu=RSOS-190412>

- Competing interests

- Authors' contributions

All submissions, other than those with a single author, must include an Authors' Contributions section which individually lists the specific contribution of each author. The list of Authors

should meet all of the following criteria; 1) substantial contributions to conception and design, or acquisition of data, or analysis and interpretation of data; 2) drafting the article or revising it critically for important intellectual content; and 3) final approval of the version to be published.

- Acknowledgements

- Funding statement

Because the schedule for publication is very tight, it is a condition of publication that you submit the revised version of your manuscript before 22-Jun-2019. Please note that the revision deadline will expire at 00.00am on this date. If you do not think you will be able to meet this date please let me know immediately.

- 1) A text file of the manuscript (tex, txt, rtf, docx or doc), references, tables (including captions) and figure captions. Do not upload a PDF as your "Main Document";

- 2) A separate electronic file of each figure (EPS or print-quality PDF preferred (either format should be produced directly from original creation package), or original software format);
- 3) Included a 100 word media summary of your paper when requested at submission. Please ensure you have entered correct contact details (email, institution and telephone) in your user account;
- 4) Included the raw data to support the claims made in your paper. You can either include your data as electronic supplementary material or upload to a repository and include the relevant doi within your manuscript. Make sure it is clear in your data accessibility statement how the data can be accessed;
- 5) All supplementary materials accompanying an accepted article will be treated as in their final form. Note that the Royal Society will neither edit nor typeset supplementary material and it will be hosted as provided. Please ensure that the supplementary material includes the paper details where possible (authors, article title, journal name).

If your manuscript is newly submitted and subsequently accepted for publication, you will be asked to pay the article processing charge, unless you request a waiver and this is approved by Royal Society Publishing. You can find out more about the charges at <http://rsos.royalsocietypublishing.org/page/charges>. Should you have any queries, please contact opscience@royalsociety.org.

Kind regards,
Alice Power
Editorial Coordinator
Royal Society Open Science
opscience@royalsociety.org

on behalf of Dr Stephen Long (Associate Editor) and Kevin Padian (Subject Editor)
opscience@royalsociety.org

Associate Editor Comments to Author (Dr Stephen Long):

This study appears novel and has potential for control of insect pests. Both reviewers recommend publication, subject to addressing some remaining points. Reviewer #1 notes the inadequacy in detailing the experimental design and also interpretation of the PCR - these must be addressed. I will review the final version of the ms for acceptability.

Reviewer comments to Author:

Reviewer: 1

Comments to the Author(s)

The manuscript titled "Chitinase genes from *Metarhizium anisopliae* for the control of Whitefly in Cotton" adds up valuable research output to plant pathology. The study appears to be authentic and may play significant role to potentially control insect pests causing spread of devastating viral diseases by employing chitinase genes from fungal strain *M. anisopliae*.

Following are minor comments to be considered,

Methodology part:

2.6.4. clarify that the number of seedlings and pots were set up in the experiment.

Results section.

Figures 2 & 3, can be supplementary files.

3.5. Expression of chitinase in cotton plants. It is obvious that the genes were detected based on end-point PCR and the data was validated by phenotypic expression. Since real expression of genes by real-time quantitative has not been done, therefore, here expression may be replaced by detection and confirmation of chitinase in cotton plants.

Reviewer: 2

Comments to the Author(s)

Dear Authors;

Whiteflies are small Hemipterans that typically feed on the undersides of cotton leaves. They comprise the family Aleyrodidae, the only family in the superfamily Aleyrodoidea. More than 1550 species have been described.

In warm or tropical climates and especially in fields, whiteflies present major problems in cotton protection. Worldwide economic losses are estimated at hundreds of millions of dollars annually for cotton business.

Therefore, the present study was designed for the harmful effects with using genetically modified technology. This is a original research with finding n results.

I am strongly suggested that "accepted with minor correction".

Author's Response to Decision Letter for (RSOS-190412.R0)

See Appendix A.

Decision letter (RSOS-190412.R1)

01-Aug-2019

Dear Dr Akhter,

I am pleased to inform you that your manuscript entitled "Chitinase genes from *Metarhizium*

anisopliae for the control of Whitefly in Cotton" is now accepted for publication in Royal Society Open Science.

Kind regards,
Andrew Dunn

Royal Society Open Science Editorial Office
Royal Society Open Science
openscience@royalsociety.org

on behalf of Dr Stephen Long (Associate Editor) and Kevin Padian (Subject Editor)
openscience@royalsociety.org

Appendix A

Nucleotide sequence has been submitted to the GenBank and following accession numbers were allotted:

Metarhizium anisopliae isolates-Tn-16: Genbank Accession No: 218605, and

Tn-25: Genbank Accession No: 218606.

All the supporting/Raw data has been uploaded in separate files as directed.

Response to Reviewer 1

1. Changes made in section 2.6.4.

The experiment comprises 5 replicates, while each replicate represent a pot with four seedlings.

2. Figure 2 and 3 has been placed in supplementary data.

3. Expression has been replaced with Detection and confirmation in the section 3.5.

Response to Reviewer 2

1. New and relevant references has been added in the introduction and discussion section.

Such as reference # 13 and 14 in the introduction, while 39, 57 and 59 number in the discussion section.

2. Marker information is added now and figure legends were also changed accordingly.

3. Corrections were made in the text according to the suggestion.

4. Conclusion is modified according to the suggestions as new sentences were added in the section.